# Foundation Reinforcement Learning: towards Embodied Generalist Agents with Foundation Prior Assistance

## Abstract

Recently, people have shown that large-scale pre-training from diverse internet-scale data is the key to building a generalist model, as witnessed in the natural language processing (NLP) area. To build an embodied generalist agent, we, as well as many other researchers, hypothesize that such foundation prior is also an indispensable component. However, it is unclear *what is the proper concrete form we should represent those embodied foundation priors* and *how those priors should be used in the downstream task*. In this paper, we focus on the concrete form in which to represent embodied foundation priors and propose an intuitive and effective set of the priors that consist of foundation policy, foundation value, and foundation success reward. The proposed priors are based on the goal-conditioned Markov decision process formulation of the task. To verify the effectiveness of the proposed priors, we instantiate an actor-critic method with the assistance of the priors, called Foundation Actor-Critic (FAC). We name our framework as **Foundation Reinforcement Learning** (FRL), since our framework completely relies on embodied foundation priors to explore, learn and reinforce. The benefits of our framework are threefold. (1) *Sample efficient learning*. With the foundation prior, FAC learns significantly faster than traditional RL. Our evaluation on the Meta-World has proved that FAC can achieve 100% success rates for 7/8 tasks under less than 200k frames, which outperforms the baseline method with careful manual-designed rewards under 1M frames. (2) *Robust to noisy priors*. Our method tolerates the unavoidable noise in embodied foundation models. We have shown that FAC works well even under heavy noise or quantization errors. (3) *Minimal human intervention*: FAC completely learns from the foundation priors, without the need of human-specified dense reward, or providing teleoperated demonstrations. Thus, FAC can be easily scaled up. We believe our FRL framework could enable the future robot to autonomously explore and learn without human intervention in the physical world. In summary, our proposed FRL framework is a novel and powerful learning paradigm, towards achieving an embodied generalist agent.

## 1 Introduction

Recently, the fields of Natural Language Processing (NLP) (Vaswani et al., 2017; Devlin et al., 2018; Brown et al., 2020; OpenAI, 2023) and Computer Vision (CV) (Dosovitskiy et al., 2020; Radford et al., 2021; Ramesh et al., 2022; Kirillov et al., 2023) have witnessed significant progress, primarily attributable to the ability to consume extensive datasets in Deep Learning (DL). Specifically, GPT models (Brown et al., 2020; OpenAI, 2023) are built upon a large pre-trained corpus consisting of billions of texts from the Internet, while Segment Anything (Kirillov et al., 2023) employs massive amounts of hand-labeled segmentation data. These large-scale models have demonstrated superior capabilities, including strong precision and generalization, by leveraging prior knowledge of common sense from substantial data (Liu et al., 2021; Sutton, 2019).

To build an embodied generalist agent, we, as well as many other researchers, believe that commonsense prior acquired from large-scale datasets is also the key. Recently, researchers have made steady progress towards this goal. A long-horizon robotic task can be solved by first decomposing the task into a sequence of primitive tasks, and then executing each primitive routine. Large language

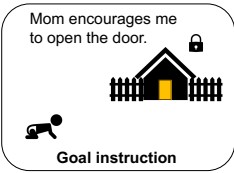 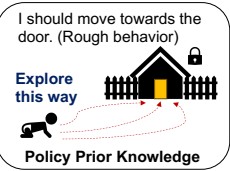 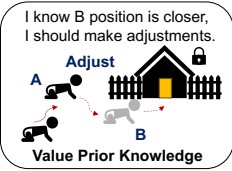 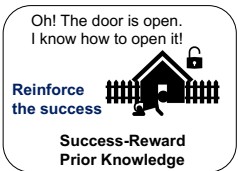

Figure 1: An example of how a child solves tasks under the three embodied prior knowledge. The proposed **Foundation Reinforcement Learning** framework follows the same learning paradigm.

models have been shown to be highly successful on the task decomposition side (Ahn et al., 2022; Liang et al., 2023; Driess et al., 2023), while the progress on the primitive skill side is still limited, which is the focus of this paper. Many recent approaches collect large amounts of human demonstrations and then use imitation learning to fine-tune the large-scale pre-trained vision language models (VLMs) (Reed et al., 2022; Brohan et al., 2022; 2023). However, these approaches do not generalize well outside the demonstration domain, and it is hard to further scale them up due to the expense of collecting demonstrations. We argue that pre-training&fine-tuning, the current wisdom of utilizing priors in LLM or VLM, might not be appropriate for embodied agents. This is because the LLMs and VLMs usually have the same output space between pre-training and fine-tuning, while in embodied applications, the action output has never been observed during pre-training. This makes it hard to generalize with the limited amount of expert demonstrations.

In this paper, we ask two fundamental questions: *What is the proper concrete form we should represent the embodied foundation prior? How should those priors be used in the downstream task?* We do not train any new foundation models, but we emphasize that our proposed framework is agnostic to any foundation prior models. We believe such meta-level questions we study above are quite significant for the following reasons. Currently, there is a large amount of research on building foundation models for embodied AI, as mentioned in related works. However, they are very different regarding the form of foundation models and are not even comparable. For example, R3M(Nair et al., 2022c) learns the visual backbone, and SayCan(Ahn et al., 2022) learns the task decomposer, while RT chooses VLM finetuning in an end-to-end way. They are studying to train the embodied foundation models from distinct perspectives, and the questions they try to answer are different. Instead, our work focuses on the concrete form in which to represent embodied foundation priors, rather than the actual RL algorithms that take advantage of the foundation priors. This is because there is no widely agreed embodied foundation model form that is widely accessible. We believe defining the form of the embodied foundation model is the first priority, which is the main contribution of this work. For example, in the BERT(Devlin et al., 2018) era, most researchers believed that BERT was the universal model, but GPT(Brown et al., 2020; OpenAI, 2023) proposed to build large language models in an autoregressive way, which is another form to represent the language foundation prior knowledge. Similarly, what our paper discusses is exactly the form of the foundation priors in embodied AI. Specifically, starting from the goal-conditioned MDP task formulation, we propose three essential prior knowledge for embodied AI: the policy, value, as well as success-reward prior. With the analogy of how humans solve tasks based on their commonsense (illustrated in Fig. 1), the proposed priors approximate the rough approach to completing it (**policy prior**), and during the execution phase, the priors can judge how good is the current state (**value prior**). The priors can also adjudicate the completion status of the task (**success-reward prior**). On top of these priors, we propose a novel framework named **Foundation Reinforcement Learning** (FRL) that runs RL with the assistance of the potentially noisy prior knowledge. To demonstrate its efficacy, we design an Actor-Critic algorithm based on the priors extracted or distilled from existing foundation models, which is called Foundational Actor-Critic (FAC).

Our FRL framework enjoys three major benefits. (1) **Sample efficient learning**. Our method explores with prior knowledge. Compared to the uninformed exploration in vanilla RL, our FAC method learns with significantly fewer samples. (2) **Robust to noisy priors**. The embodied priors can be noisy, since they are pre-trained from other embodiment data. Our FAC method works robustly even under heavily noised policy priors. (3) **Minimal human intervention**: FAC completely learns from the foundation priors, without the need of human-specified rewards or providing tele-operated demonstrations. This starkly contrasts some previous work that heavily relies on large-scale demonstrations (Reed et al., 2022; Brohan et al., 2022; 2023). And we do not limit the way to acquiring corresponding foundation priors. Thus, FAC can potentially be scaled up more easily.

We apply the FAC to robotics tasks in simulation, and empirical results have shown the strong performance of FAC. In summary, our contributions are as follows:

- We propose the Foundation Reinforcement Learning (FRL) framework. The framework systematically introduces three priors that are essential to embodied agents. Our framework also suggests how to utilize those priors with RL, which is proven to be highly effective.

- We propose the Foundational Actor-Critic (FAC) algorithm, a concrete algorithm under the FRL framework that utilizes the policy, value, and success-reward prior knowledge. FAC is sample efficient; it can learn with noisy priors; it requires minimal human effort to run.

- Empirical results show that FAC can achieve 100% success rates in 7/8 tasks under 200k frames without manual-designed rewards, which proves the high sample efficiency. The ablation experiments verify the significance of the embodied foundation prior and the robustness of our proposed method w.r.t. the quality of the foundation prior.

## 2 RELATED WORK

**Foundation Models for Policy Learning** The ability to leverage generalized knowledge from large and varied datasets has been proved in the fields of CV and NLP. In embodied AI, researchers attempt to learn universal policies based on large language models (LLMs) or vision-language models (VLMs). Some researchers train large transformers by tokenizing inputs and inferring actions by imitation learning (Brohan et al., 2022; 2023; Reed et al., 2022; Yu et al., 2023), or offline Reinforcement Learning (Chebotar et al., 2023). Some researchers utilize the LLMs as reasoning tools and do low-level control based on language descriptions (Di Palo et al.; Huang et al., 2022; Ahn et al., 2022; Driess et al., 2023; Wu et al., 2023; Singh et al., 2023; Shridhar et al., 2022). The above works utilize human teleoperation to collect data for policy learning. However, it is hard to scale up human teleoperation to collect large-scale data. The model UniPi (Du et al., 2023) predicts videos for tasks based on VLMs and generates actions via a trained inverse model from the videos. But UniPi is of poor robustness empirically due to the lack of interactions with the environments.

**Foundation Models for Representation Learning** Apart from learning policies directly from the foundation models, some researchers attempt to extract universal representations for downstream tasks. Some works focus on pre-trained visual representations that initialize the perception encoder or extract latent states of image inputs (Karamcheti et al., 2023; Shah & Kumar, 2021; Majumdar et al., 2023; Nair et al., 2022c). Some researchers incorporate the pre-trained LLMs or VLMs for linguistic instruction encoding (Shridhar et al., 2023; Nair et al., 2022b; Jiang et al., 2022). Some have investigated how to apply the LLMs or VLMs for universal reward or value representation in RL. Fan et al. (2022); Nair et al. (2022a); Mahmoudieh et al. (2022) build language-conditioned reward foundation models to generate task reward signals, and Ma et al. (2022) is the first to train a universal goal-conditioned value function on large-scale unlabeled videos. However, no policy prior knowledge is provided for down-steam policy learning in these methods, and we find it significant of the policy prior knowledge for down-steam tasks in some experiments. In contrast, our proposed framework FRL leverages policy, value, and success-reward prior knowledge, which covers the basic commonsense of solving sequential tasks.

## 3 BACKGROUND

### 3.1 ACTOR-CRITIC ALGORITHMS

Since various Actor-Critic algorithms demonstrate great performance on diverse tasks (Haarnoja et al., 2018; Lillicrap et al., 2015), we build our method on top of Actor-Critic algorithms to demonstrate the Foundation Reinforcement Learning framework, which is named Foundation Actor-Critic. Specifically, we choose a variant of deterministic Actor-Critic algorithms DrQ-v2 as the baseline, which is a SoTA model-free method for visual RL. It learns Q-value functions with clipped double Q-learning (Fujimoto et al., 2018) and deterministic policies by the Deterministic Policy Gradient (DPG) (Silver et al., 2014), which maximizes $J_\phi(\mathcal{D}) = \mathbb{E}_{s_t \sim \mathcal{D}}[Q_\theta(s_t, \pi_\phi(s_t)]$. Here $\mathcal{D}$ is the dataset of the training replay, and $\theta, \phi$ are the learnable parameters. The training objectives of DrQ-v2 are as follows (Yarats et al., 2021):

$$\mathcal{L}_{\text{actor}}(\phi) = -\mathbb{E}_{s_t \sim \mathcal{D}} \left[ \min_{k=1,2} Q_{\theta_k}(s_t, a_t) \right], k \in \{1, 2\};$$

$$\mathcal{L}_{\text{critic}}(\theta) = \mathbb{E}_{s_t \sim \mathcal{D}} \left[ (Q_{\theta_k}(s_t, a_t) - y)^2 \right], k \in \{1, 2\};$$

(1)

where $s_t$ is the latent state representation, $a_t$ is the action sampled from actor $\pi_\phi$, and $y$ is the n-step TD target value. More details can be referred to Yarats et al. (2021).

## 3.2 Reward shaping in MDP

In this work, we apply the value prior knowledge in Actor-Critic algorithms in the format of the reward shaping. Reward shaping guides the RL process of an agent by supplying additional rewards for the MDP (Dorigo & Colombetti, 1998; Mataric, 1994; Randløv & Alstrøm, 1998). In practice, it is considered a promising method to speed up the learning process for complex problems. Ng et al. (1999) introduce a formal framework for designing shaping rewards. Specifically, we define the MDP $\mathcal{G} = (\mathcal{S}, \mathcal{A}, \mathcal{P}, \mathcal{R})$, where $\mathcal{A}$ denotes the action space, and $\mathcal{P} = \Pr\{s_{t+1}|s_t, a_t\}$ denotes the transition probabilities. Rather than handling the MDP $\mathcal{G}$, the agent learns policies on some transformed MDP $\mathcal{G}' = (\mathcal{S}, \mathcal{A}, \mathcal{P}, \mathcal{R}')$, $\mathcal{R}' = \mathcal{R} + F$, where $F$ is the shaping reward function. When there exists a state-only function $\Phi : \mathcal{S} \to \mathbb{R}^1$ such that $F(s, a, s') = \gamma\Phi(s') - \Phi(s)$ ($\gamma$ is the discounting factor), the $F$ is called a **potential-based shaping function**. Ng et al. (1999) prove that the potential-based shaping function $F$ has optimal policy consistency under some conditions (Theorem 1 in the App. A.4).

The theorem indicates that potential-based function $F$ exhibits no particular preference for any policy other than the optimal policy $\pi^*_\mathcal{G}$ when switching from $\mathcal{G}$ to $\mathcal{G}'$. Moreover, under the guidance of shaping rewards for the agents, a significant reduction in learning time can be achieved. In practical settings, the real-valued function $\Phi$ can be determined based on domain knowledge.

## 4 Method

In this section, we investigate what kinds of prior knowledge are significant for training embodied generalist agents and how to leverage the prior knowledge to the given down-steam tasks.

### 4.1 Foundation Prior Knowledge in Embodied AI

For embodied intelligent agents, the process of handling different tasks in environments can be formulated as **goal-conditioned MDP** (GCMDP) $\mathcal{G}$: $\mathcal{G} = (\mathcal{S}, \mathcal{A}, \mathcal{P}, \mathcal{R}_{|\mathcal{T}}, \mathcal{T})$. $\mathcal{S} \in \mathbb{R}^m$ denotes the state. $\mathcal{T}$ is the task identifier. $\mathcal{R}_{|\mathcal{T}}$ denotes the rewards conditioned on tasks, which is a 0-1 success signal reward. Here, we take an example of how children solve daily manipulation tasks with commonsense prior knowledge and propose the priors in GCMDP correspondingly, as shown in Fig. 1. As a 3-year-old child, Alice has never opened a door, and today, she is encouraged to open a door. Alice receives the task instruction, and she begins to make attempts. Here, the language instruction is the description of the **goal** in the GCMDP. As a child, she has witnessed how her parents opened the door, and she has some commonsense about the task.

First, she has noticed some rough behaviors of reaching the door and turning the doorknob to open it from her parents. So she can follow their behavior and make attempts. In MDP, the commonsense of the rough behavior can be formulated as a goal-conditioned policy function, $M_\pi(s|\mathcal{T}) : \mathcal{S} \times \mathcal{T} \to \mathcal{A}$, which provides the rough action in the given task. We define such commonsense as the **policy prior knowledge**, guiding the agents with noisy actions to explore more efficiently.

She recognizes that states nearer the door are more likely to lead to success. If she encounters unfavorable states, she understands the necessity to adjust back to a more disarable one. In MDP, such commonsense can be formulated as a goal-conditioned value function $M_\mathcal{V}(s|\mathcal{T}) : \mathcal{S} \times \mathcal{T} \to \mathbb{R}^1$, which provides the value estimations of states concerning the given task. We define such commonsense as the **value prior knowledge**, measuring the values of states from the foundation models.

After several attempts, Alice observes the door is open and reinforces the behavior from her successful attempt, enabling her to consistently solve the task. Humans naturally recognize success and adjust their actions accordingly. In MDP, such commonsense can be formulated as the 0-1 success-reward function $M_\mathcal{R}(s|\mathcal{T}) : \mathcal{S} \times \mathcal{T} \to \{0, 1\}$, which equals 1 only if the task succeeds. This approach allows Alice to make trials and succeed in new tasks under the same learning pattern.

Drawing inspiration from the example, we believe the three prior knowledge are fundamental for versatile embodied agents. Consequently, following the learning paradigm, we formulate a novel

framework **Foundation Reinforcement Learning** (FRL) to solve the GCMDP, which does RL under noisy prior knowledge from foundation models. For convenience, we note such prior knowledge acquired from foundation models as the embodied foundation prior, i.e. the embodied foundation prior = {policy prior, value prior, success-reward prior}. The embodiment can be facilitated by the above three embodied prior knowledge to solve the MDP. Notably, for humans, determining whether a task has been completed is precise and straightforward, but judging how good the current state is and where to take action can be vague and noisy. Thus, we assume the success-reward prior knowledge is relatively precise and sound, but the value and policy prior knowledge can be much more noisy. And we also make ablations to investigate how the performance of FAC can be affected by the quality of the priors in Sec. 5.3.

To sum up, compared to the setting of vanilla RL, all the signals for the Foundation RL come from the foundation models. The vanilla RL relies on uninformative trial and error explorations as well as carefully and manually designed reward functions to learn the desired behaviors. It is not only of poor sample efficiency but also requires lots of human reward engineering. Instead, in Foundation RL, $M_\mathcal{V}$ and $M_\mathcal{R}$ give the value and reward estimations of states, and $M_\pi$ provides behavioral guidance with rough prior actions for the agent. This way, it can solve tasks much more efficiently, and learn with minimal human intervention.

## 4.2 FOUNDATION ACTOR-CRITIC

Under the proposed Foundation RL framework, we instantiate an actor-critic algorithm to take advantage of the prior knowledge, which is named Foundation Actor-Critic (FAC). In this work, we systematically demonstrate how to inject the policy, value, and success-reward embodied prior knowledge into Actor-Critic algorithms, but do not limit how to acquire and leverage them, indicating that other approaches may be possible. The policy prior informs how to act at each step, while the value prior can correct the policy when it enters bad states. The success-reward prior tells the agent whether the task is successful to reinforce the successful experiences.

**Formulated as 0-1 Success-reward MDP** Generally, as we mention above, we consider the given task as an MDP with 0-1 success-rewards from foundation model $M_\mathcal{R}$. Thus, we name the MDP to solve as $\mathcal{G}_1$, where $\mathcal{R}_{\mathcal{G}_1|\mathcal{T}} = M_\mathcal{R}(s|\mathcal{T}) \in \{0, 1\}$. However, the MDP problem is in sequence, and it is difficult to optimize the policy based on the 0-1 sparse reward signal due to the large search space of the MDP. We utilize the policy and value prior knowledge to learn $\mathcal{G}_1$ better and more efficiently.

**Guided by Policy Regularization** To encourage the embodiment to explore the environments with the guidance of policy prior knowledge, we regularize the actor $\pi_\phi$ by the policy prior from $M_\pi(s|\mathcal{T})$ during training. Such regularization can help the actor explore the environment. Without loss of generality, we assume that the policy foundation prior follows Gaussian distributions. We add a regularization term to the actor: $\mathcal{L}_{\text{reg}}(\phi) = \text{KL}(\pi_\phi, \mathcal{N}(M_\pi(s_t|\mathcal{T}), \hat{\sigma}^2))$, where $\hat{\sigma}$ is the standard deviation hyper-parameter of the policy prior. Such regularization item is simple to implement but effective, which is widely used in other algorithms (Haldar et al., 2023; Lancaster et al., 2023). Note that there might be some bias caused by the policy prior; however, the bias can be bounded theoretically, as shown in Appendix (Theorem 2).

**Guided by Reward-shaping from Value Prior** The noisy foundation policy prior might mislead the agent to undesired states. We propose to guide the policy by the value model $M_\mathcal{V}(s|\mathcal{T})$ to avoid unnecessary exploration of unpromising states. Since there is a value function in the Actor-Critic algorithm, a natural approach is initializing with $M_\mathcal{V}(s|\mathcal{T})$ and fine-tuning the value functions. However, we empirically find that it performs poorly due to the catastrophic forgetting issues.

To better employ the value foundation prior, we propose to utilize the **reward-shaping** technique (Ng et al., 1999). Specifically, we introduce the potential-based shaping function $F(s, s'|\mathcal{T}) = \gamma M_\mathcal{V}(s'|\mathcal{T}) - M_\mathcal{V}(s|\mathcal{T})$. Intuitively, since $M_\mathcal{V}$ roughly estimates the value of each state, $F$ can measure the increase of value towards reaching the next state $s'$ from the current state $s$. The shaping reward will be positive if the new state $s'$ is better than the state $s$.

**Foundation Actor-Critic** In summary, we propose to deal with a new MDP $\mathcal{G}_2$, where $\mathcal{R}_{\mathcal{G}_2|\mathcal{T}} = \alpha M_\mathcal{R} + F, \alpha > 0$. Here, the $\alpha$ emphasizes the success feedback, which equals 100 in this work. The optimal policy of $\mathcal{G}_2$ is the same as that of $\mathcal{G}_1$ because $F$ is a potential-based function. In this paper, we build our model upon the DrQ-v2 (Yarats et al., 2021) with image inputs, where the objectives

are listed in Eq. (1). We inject the value, success-reward, and policy foundation prior to the baseline method to solve $\mathcal{G}_2$. We name the proposed foundation RL method as **Foundation Actor-Critic** (FAC). The objective of critics in FAC remains the same as Eq.(1), and the objective of the actor in FAC is regularized, listed in Eq. (2).

$$\mathcal{L}_{\text{actor}}(\phi) = -\mathbb{E}_{s_t \sim \mathcal{D}} \left[ \min_{k=1,2} Q_{\theta_k}(s_t, a_t) \right] + \beta \text{KL}(\pi_\phi, \mathcal{N}(M_\pi(s_t | \mathcal{T}), \hat{\sigma}^2))$$

$$r_t = \alpha M_{\mathcal{R}}(s_t | \mathcal{T}) + \gamma M_{\mathcal{V}}(s_{t+1} | \mathcal{T}) - M_{\mathcal{V}}(s_t | \mathcal{T})$$

(2)

, where $\beta$ is the tradeoff between the policy gradient guidance and the policy regularization guidance, which is set to 1. FAC learns from the foundation success prior, with foundation value shaping. Thus, we don't need to manually design a reward function.

### 4.3 Acquiring Foundation Prior in FAC

In this work, we aim to study what kind of prior is important for the embodied generalist agent and how to use those priors. Building the large-scale foundation priors is out of our paper's scope. However, we think it is an exciting future research direction. To validate our proposed framework, we utilize several existing works as proxy embodied foundation models.

**Value Foundation Prior** For the value foundation model $M_{\mathcal{V}}(s_t | \mathcal{T})$, we propose to utilize VIP (Ma et al., 2022), which trains a universal value function via pre-training on internet-scale datasets. For each task, it requires a goal image and measures the distance between the current state and the goal state visually. In the experiments, we load the pre-trained value foundation model and apply it directly to value inference without in-domain fine-tuning.

**Policy Foundation Prior** For the policy foundation model $M_\pi(s_t | \mathcal{T})$, we propose to follow the work UniPi, which infers actions based on a language conditioned video prediction model and a universal inverse dynamics model. UniPi (Du et al., 2023) generates video trajectory for the task $\mathcal{T}$ conditioned on the start frame and task. Then, actions can be extracted from the generated video with a universal inverse dynamics model. However, video generation is computationally expensive. To improve the computational efficiency, we propose to distill a policy prior model $M_\pi(s_t | \mathcal{T})$ from the video generation model and the inverse dynamics model. Specifically, we generate video datasets from the video generation model and label the corresponding actions from the video dataset by the inverse model. Then, we train a policy model $M_\pi(s_t | \mathcal{T})$ from the datasets by supervised learning, which takes the current state as input. The original UniPi performs heavy in-domain fine-tuning. Instead, we use a few data for in-domain fine-tuning, which is more practical for real-world applications. More details are attached in the App. A.3.

**Reward Foundation Prior** There are few foundation models to distinguish the success behavior in embodied AI. Therefore, we use the ground truth 0-1 success signal from the environments. Additionally, we also build a success-reward model $M_{\mathcal{R}}(s_t | \mathcal{T})$ to provide the success signals, which indicates no signals come from the environment. Specifically, we distill a success-reward model from 50k ground-truth replay data for the 8 tasks in total. We conduct ablation studies towards the quality of the reward model in App. A.1.

## 5 Experiments

In this section, we provide extensive evaluations of Foundation Actor-Critic on robotics manipulation tasks. We attempt to investigate the effects of the foundation prior knowledge in policy learning, especially the sample efficiency and robustness aspects. Specifically, our experiments are designed to answer the following questions: (a) How sample efficient is our proposed Foundation Actor-Critic algorithm; (b) How significant are the three foundation prior knowledge respectively; (c) How is the quality of the foundation model affect the performance of FAC.

### 5.1 Setup

**Building Policy Foundation Models in FAC** As mentioned in Sec. 4.3, we distill a policy foundation model $M_\pi$ through a language condition video prediction model and an inverse dynamics model $\rho(s_t, s_{t+1})$. As for the video prediction model, we choose the open-source vision language

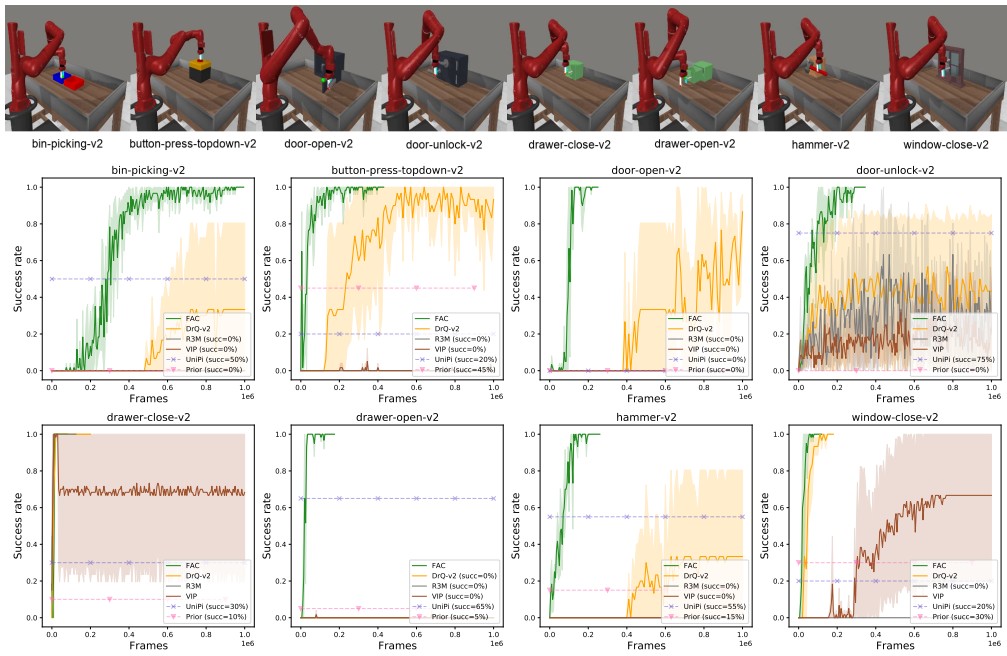

Figure 2: Success rate curves for the 8 tasks in Meta-World. Our FAC can achieve **100% success rates** for all tasks under the limited performance of the policy prior model. In FAC, 7/8 tasks can be solved at 100% rate less than **200k** frames, which significantly outperforms the baselines DrQ-v2 and R3M with manual-designed rewards.

model, Seer (Gu et al., 2023). The model Seer predicts a video conditioned on one image and a language instruction with latent diffusion models, pre-trained on Something Something V2 (Goyal et al., 2017) and BridgeData (Ebert et al., 2021) datasets. Ideally, the model can be plugged in without in-domain fine-tuning. However, we find the current open-source video prediction models fail to generate reasonable videos in the simulator. Consequently, we fine-tune the Seer with 10 example videos from each task. Compared to UniPi that fine-tunes with 200k videos, the videos generated by our model are more noisy. Our model reflects the noisy nature of the future policy foundation model. In terms of the inverse dynamics model $\rho(s_t, s_{t+1})$, we save the replay buffer of the baseline DrQ-v2, containing 1M frames of each task, and train $\rho(s_t, s_{t+1})$ based on the replay buffers. Finally, to distill the policy foundation model $M_\pi$, we generate 100 videos for each task from the fine-tuned Seer model (1k videos for the harder task bin-picking), label pseudo actions among the videos by the inverse dynamics model $\rho(s_t, s_{t+1})$, and train $M_\pi(s_t|\mathcal{T})$ on the action-labeled videos by supervised learning. More details are attached in the App. A.3.

**Environments and Baselines** We conduct experiments of FAC on 8 tasks from simulated robotics environments Meta-World (Yu et al., 2020), which are commonly used because they test different manipulation skills (Haldar et al., 2023). We average the success rates over 20 evaluation episodes across 3 runs with different seeds. To verify the effectiveness and significance of the three foundation priors, we compare our methods to the following baselines: (**1**) Vanilla DrQ-v2 (Yarats et al., 2021), with manually designed rewards from the suite; (**2**) R3M (Nair et al., 2022c), VIP (Ma et al., 2022). We combine DrQ-v2 with the R3M visual representation or VIP visual representation. Same as the vanilla DrQ-v2, this baseline also learns from manually designed rewards; (**3**) UniPi (Du et al., 2023), which infers actions by the inverse dynamics model $\rho(s_t, s_{t+1})$ and an expert video generated from the language conditioned video prediction model Seer, fine-tuned under 10 example videos for each task following our setups; (**4**) The distilled policy foundation model $M_\pi(s_t|\mathcal{T})$.

## 5.2 PERFORMANCE ANALYSIS

We compare the performance of our method with the above baselines on 8 tasks in Meta-World with 1M frames. Our proposed FAC achieves 100% success rates for all the tasks. 7/8 of them require less than 200k frames. For the hard task bin-picking, FAC requires less than 400k frames. However, the baseline methods can not achieve 100% success rates on most tasks. As illustrated

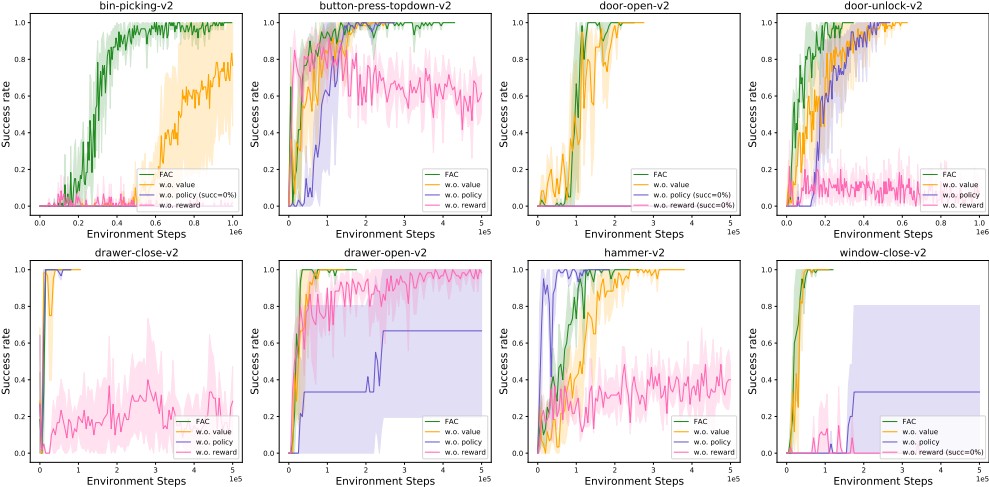

Figure 3: Ablation of the three embodied prior knowledge. The success-reward prior is the most significant. The policy prior is necessary for hard tasks, and the value prior makes the learning process more sample efficient.

in Fig. 2, the sample efficiency and the success rates of FAC are much superior compared to the baseline methods. DrQ-v2 is able to complete some tasks but learns much slower compared to FAC. R3M and VIP backbones inject the visual representation prior knowledge into the RL process, but their performance are even worse than DrQ-v2. We hypothesize that it might be caused by the pre-trained model having lost plasticity (D'Oro et al., 2022). Since the UniPi and the distilled foundation prior baseline do not involve training, they are represented as two horizontal lines in Fig. 2. UniPi outperforms the distilled prior in most environments, since the distilled prior is learned from UniPi. However, UniPi is still far inferior compared to FAC.

## 5.3 ABLATION STUDY

In this section, we answer the following questions: (a) Are all the three proposed priors necessary? What's the importance of each? (b) How does FAC perform with better / worse foundation priors?

**Ablation of Each Embodied Foundation Prior** To investigate the importance of each foundation prior, we remove each prior and compare it against the full method. Figure 3 shows the three ablations: no policy prior (i.e. no policy KL regularization), no value prior (i.e. $\mathcal{R}_{|\mathcal{T}} = \alpha M_{\mathcal{R}}(s|\mathcal{T})$), no success reward (i.e. $\mathcal{R}_{|\mathcal{T}} = \gamma M_{\mathcal{V}}(s_{t+1}|\mathcal{T}) - M_{\mathcal{V}}(s_t|\mathcal{T})$ ).

We find that the reward prior is the most important, without which the performance over all the tasks drops a lot. The reason is that, without the 0-1 success reward signals, the reward function is only a shaping reward, where any policy is equivalent under this reward.

Without the policy prior, the agent fails on some hard

with policy prior     w.o. policy prior

Figure 4: Both succeed. With policy guidance, the agent uses the hammer to nail. Without policy guidance, the agent uses the gripper to nail.

tasks, such as bin-picking and door-open. It also converges much more slowly on drawer-open and window-close. We note that the task hammer converges faster without the policy prior, which is counter-intuitive. This is because the agent w.o. policy prior succeeds through pushing the nail with the robot arm rather than with the hammer, as illustrated in Fig. 4.

Without the value prior, the sample efficiency would drop, especially for the hard task bin-picking. Under the noisy policy prior, the shaping rewards inferred from value prior can guide the policy to reach states of higher values with larger probability. Generally, applying all the foundation priors for most environments will be best. However, in some environments, either the policy prior or the value prior is accurate enough for solving the tasks, resulting in a few performance drops when removing

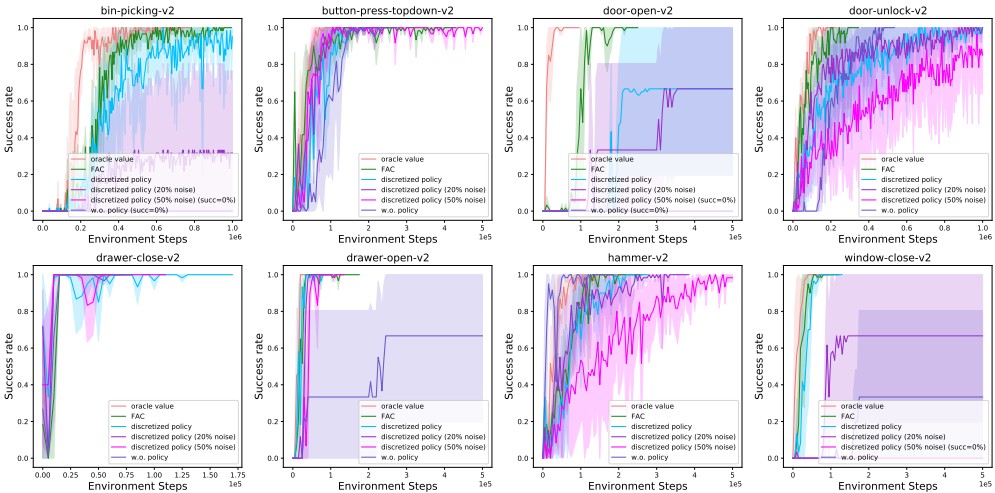

Figure 5: Ablation of the quality of the value and policy prior knowledge. We observe that (1) better prior knowledge leads to better FAC performance; (2) FAC is robust to the quality of the foundation priors. FAC works with 20% or even 50% noisy discretized policy prior.

the other prior, such as button-press-topdown and door-open. This depends on the quality of the foundation prior conditioned on the tasks. Nevertheless, learning from the three embodied prior can be better than that from only two of them in the 8 environments.

**FAC with Various Quality of Value and Policy Foundation Prior** Since the value foundation prior is from VIP (Ma et al., 2022) without in-domain data fine-tuning, which can be noisy, we are interested in how the noisy values perform compared to the ground truth values. We build oracle value functions for each task from the pre-trained FAC models. We find that oracle values give a further boost to some tasks (Figure 5), such as bin-picking-v2 and door-open-v2. This indicates that better value foundation priors might further boost the performance.

We have shown that although the distilled policy prior itself has low success rates on most tasks (Fig. 2), FAC is able to achieve high success rates by utilizing the noisy policy prior. To further test the robustness of our method, we define several noisier policy priors. Specifically, we discretize each action dimension generated from the distilled policy model into only three values $\{-1, 0, +1\}$. This makes the policy prior only contain rough directional information. We name this prior as *discritized policy*. To generate even noisier prior, we replace the discretized actions with uniform noise at 20% and 50% probability. As shown in Figure 5, we find that the discretized policy prior performs similarly to the original policy prior, except for door-open. Adding the extra uniform noise decreases the performance further. However, we note that even using the discretized policy with 50% noise, FAC can still reach 100% success rates in many environments. We also ablate the quality of the success-reward foundation prior in App. A.1. In conclusion, the results indicate that FAC is robust to the quality of the foundation prior. The better the prior is, the more sample-efficient FAC is.

## 6 DISCUSSION

In this paper, we introduce a novel framework, termed Foundation Reinforcement Learning, which leverages policy, value, and success-reward prior knowledge for reinforcement learning tasks. Additionally, we elucidate the application of the embodied foundation prior within actor-critic methodologies, hereby designated as Foundation Actor-Critic. We acknowledge there are some limitations in this work. More comparisons can be made with other methodologies like VLM that have a noted advantage in generating language-instructed policies. And to make the foundation prior not messy, we finetune or distill foundation models with small amounts of data. Moreover, We acknowledge that there are two dimen- sions for future exploration of this work. For one thing, it is imperative to construct accurate and broadly applicable foundation priors, which is out of our scope. For another, it is promising to introduce more abundant prior knowledge for the Foundation RL. For instance, humans can predict the future states. Such prediction prior knowledge can be extracted from the dynamic foundation models, which can be potentially effective for policy learning.

## 7 REPRODUCIBILITY STATEMENT

The main implementations of our proposed method are in Sec. 4.2. The details of designing and training foundation prior models are in Sec. 4.3 and App. A.3. In addition, the settings of the experiments and hyper-parameters we choose are in App. A.3.

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

# A   APPENDIX

## A.1   MORE ABLATION RESULTS

**FAC with Various Quality of Success-reward Foundation Prior** In the previous experiments, we assume that the success-reward signals come from the environment. It is necessary to conduct experiments with success-reward foundation prior. Considering there are few universal success-reward foundation models, we distill one proxy success-reward model with 50k data in total for all 8 tasks, which are labeled by the ground-truth success signals from replay buffers. The proxy model takes images as input and is conditioned on the task embeddings (multi-task), which has 1.7% false positive error and 9.9% false negative error on the evaluation datasets. Then, we run FAC with the three priors without any signals from the environment during training.

The results are attached in Fig. 6 in App.A.1 (Page 14). We find that compared to receiving the ground-truth success-reward signals, FAC under the 50k-images-distilled model has a limited performance drop in the tasks generally. And it can achieve much superior performance than the FAC w.o. reward. Consequently, the proposed FAC can work well under the noisy success-reward foundation prior. It gives the potential that we can use foundation model in place of human-specified reward for any new tasks.

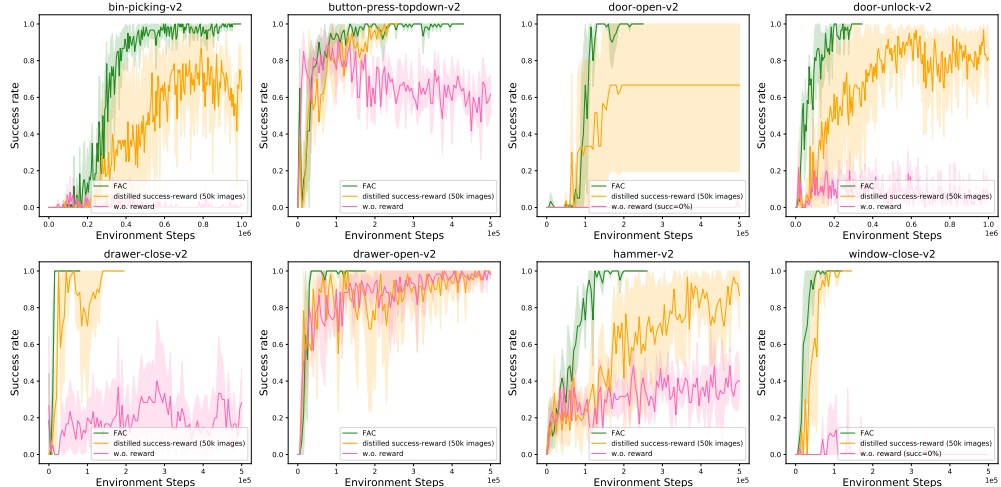

Figure 6: Ablation of the quality of the success-reward prior knowledge. The 50k-image-distilled success-reward model has 1.7% false positive error and 9.9% false negative error. The FAC can work well under the noisy success-reward foundation prior.

## A.2   EXPERIMENTAL RESULTS WITH MORE BASELINES

**Comparison to More Baselines with Success-reward Only** Here, we also add some baselines under the setting, where only the success-reward foundation prior is provided. We choose the recent SOTA model-free RL algorithms on MetaWorld ALIX (Cetin et al., 2022) and TACO (Zheng et al., 2023), as well as the baseline DrQ-v2 (Yarats et al., 2021) with the success-reward only. Notably, ALIX and TACO are both built on DrQ-v2. The results are shown in Fig. 7, where '*' means that only 0-1 success reward is given. Only FAC can achieve 100% success rates in all the environments. DrQ-v2*, ALIX*, TACO* can not work on hard tasks such as bin-picking and door-open. And FAC requires fewer environmental steps to reach 100% success rates, as shown in the Figure. The results on the new baselines can verify the significance and efficiency of utilizing the abundant prior knowledge for RL in a way.

## A.3   IMPLEMENTATION DETAILS

Since FAC is built on top of DrQ-v2, the hyper-parameters of training the actor-critic model are the same as DrQ-v2 (Yarats et al., 2021). The n-step TD target value and the action in Eq. 2 are as

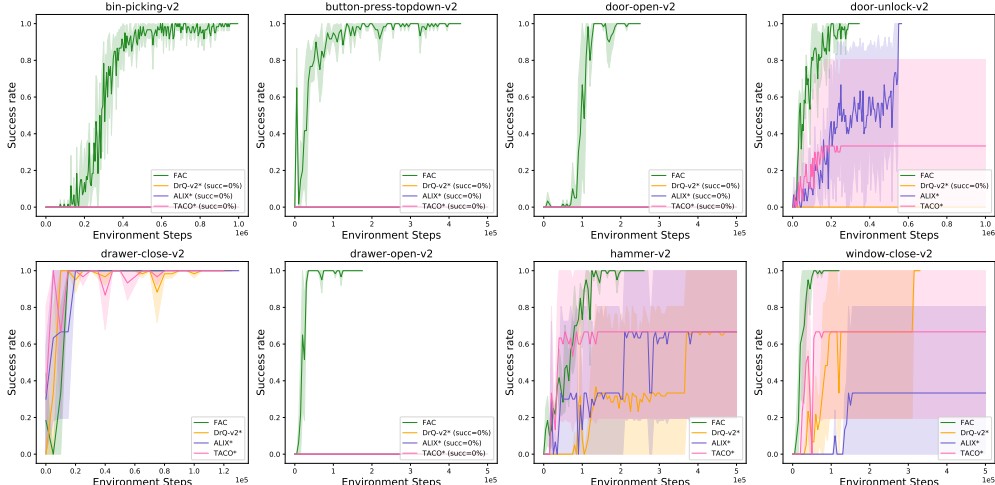

Figure 7: Here '*' in DrQ-v2, ALIX and TACO means only the 0-1 success reward is provided from the environment, which is different from the original settings in their works. FAC can work for all the tasks while the other baselines fail in half of them. It is significant and sample-efficient to utilize prior knowledge for reinforcement learning.

follows, where $\bar{\theta}_k$ are the moving weights for the Q target networks.

$$y = \sum_{i=0}^{n-1} \gamma^i r_{t+i} + \gamma^n \min_{k=1,2} Q_{\bar{\theta}_k}(s_{t+n}, a_{t+n}),$$

$$a_t = \pi_\phi(s_t) + \epsilon, \epsilon \sim \text{clip}(\mathcal{N}(0, \sigma^2), -c, c) \tag{3}$$

Morever, the observation shape is $84 \times 84$, and we stack 3 frames while repeating actions for 2 steps. In Meta-world, we follow the experimental setup of (Haldar et al., 2023). Specifically, the horizon length is set to 125 frames for all tasks except for bin-picking and button-press-topdown, which are set to 175. Notably, we set the same camera view of all the tasks for consistency. As for the policy regularization term, the KL objective can be simplified to the MSE objective, which indicates the implemented training objective of the actor is:

$$\mathcal{L}_{\text{actor}}(\phi) = -\mathbb{E}_{s_t \sim \mathcal{D}} \left[ \min_{k=1,2} Q_{\theta_k}(s_t, a_t) \right] + \beta \left\| a_t - M_\pi(s_t | \mathcal{T}) \right\|^2, a_t \sim \pi_\phi(s_t).$$

$$r_t = \alpha M_\mathcal{R}(s_t | \mathcal{T}) + \gamma M_\mathcal{V}(s_{t+1} | \mathcal{T}) - M_\mathcal{V}(s_t | \mathcal{T}). \tag{4}$$

**Training Inverse Dynamics Model** We build the inverse dynamics model $\rho(s_t, s_{t+1})$ as follows:

- Takes inputs as $s_t, s_{t+1}$, with the shape of $3 \times 84 \times 84$.
- A Downsample Model, which outputs the representation with the shape of $128 \times 2 \times 2$.
- Flatten the planes into 512-dimension vectors.
- 1 Linear layer with ReLU, which outputs the 64-dimension vectors.
- 1 Linear layer with ReLU, which outputs the 64-dimension vectors.
- 1 Linear layer with ReLU, which outputs the action dimension vectors (equal to 4).

The Downsample model is designed as follows:

- 1 convolution with stride 2 and 128 output planes, output resolution $42 \times 42$. (ReLU)
- 2 residual block with 128 planes.
- Average pooling with stride 2 (kernel size is 3), output resolution $21 \times 21$. (ReLU)
- 2 residual block with 128 planes.
- Average pooling with stride 3 (kernel size is 5), output resolution $7 \times 7$. (ReLU)

- 2 residual block with 128 planes.
- Average pooling with stride 3 (kernel size is 4, no padding), output resolution $2 \times 2$. (ReLU)

We use the 1M replay buffer trained from vanilla DrQ-v2 for each task and collect them together as the dataset.

**Distilling Policy Foundation Models** We use the fine-tuned VLM Seer to collect 100 videos for each task (1000 in bin-picking-v2), and use the trained inverse dynamics model $\rho(s_t, s_{t+1})$ to label pseudo actions for the videos. Then, we do supervised learning to train the policy foundation prior model under the dataset, which is conditioned on the task. For convenience, we encode the task embedding as a one-hot vector, which labels the corresponding task. Thus, the size of the task embedding is 8. Here, the architecture of the distilled policy model is as follows, where the downsample model is the same as that in the inverse dynamics model.

- Takes inputs as $s_t, e_t$, with the shape of $3 \times 84 \times 84$ and $1 \times 8$.
- A Downsample Model, which outputs the representation with the shape of $128 \times 2 \times 2$.
- Flatten the planes into 512-dimension vectors.
- Concat the 512 vector and the task embedding into 520-dimension vectors.
- 1 Linear layer with ReLU, which outputs the 64-dimension vectors.
- 1 Linear layer with ReLU, which outputs the 64-dimension vectors.
- 1 Linear layer with ReLU, which outputs the action dimension vectors (equal to 4).

The training hyper-parameters of the inverse dynamics model $\rho(s_t, s_{t+1})$ and the distilled policy model $M_\pi(s_t|\mathcal{T})$ are in Table 1. The hyper-parameters of training FAC agents are the same as DrQ-v2 (Yarats et al., 2021).

Table 1: Hyper-parameters for Building the Policy Foundation Models in FAC.

| Parameter | Training $\rho(s_t, s_{t+1})$ | Training $M_\pi(s_t|\mathcal{T})$ |
|---|---|---|
| Minibatch size | 256 | 256 |
| Optimizer | AdamW | AdamW |
| Optimizer: learning rate | 1e-4 | 5e-4 |
| Optimizer: weight decay | 1e-4 | 1e-4 |
| Learning rate schedule | Cosine | Cosine |
| Max gradient norm | 1 | 1 |
| Training Epochs | 50 | 300 |

## A.4 OPTIMALITY OF POTENTIAL-BASED SHAPING FUNCTION

**Theorem 1** *(Ng et al., 1999) Suppose that $F$ takes the form of $F(s, a, s') = \gamma\Phi(s') - \Phi(s)$, $\Phi(s_0) = 0$ if $\gamma = 1$, then for $\forall s \in \mathcal{S}, a \in \mathcal{A}$, the potential-based $F$ preserve optimal policies and we have:*

$$Q_{\mathcal{G}'}^*(s, a) = Q_{\mathcal{G}}^*(s, a) - \Phi(s)$$
$$V_{\mathcal{G}'}^*(s) = V_{\mathcal{G}}^*(s) - \Phi(s) \tag{5}$$

## A.5 PROOF OF THE OPTIMALITY UNDER POLICY REGULARIZATION

**Lemma 1** *The policy $\pi_m = \frac{1}{1+\beta}\hat{\pi}_{\phi_m} + \frac{\beta}{1+\beta}M_\pi$, is the solution to the optimization problem of the actor shown in Equation 2.*

**Proof 1** *First, $\hat{\pi}_{\phi_m}$ is the RL policy optimized by standard RL optimization problem in m-th iteration, illustrated in the following equation.*

$$\hat{\pi}_{\phi_m} = \arg\max_{\hat{\pi}_\phi} \mathbb{E}_{\tau \sim \hat{\pi}_\phi}[Q(s, a)] \quad as\ m \to \infty \tag{6}$$

*Note that the following derivation omits the variance of Gaussian distribution for convenience. This is because the variance is independent of the state in the deterministic Actor-Critic algorithms DrQ-v2 algorithm.*

*According to Equation 2, the policy $\pi_m$ can be represented as:*

$$\pi_m = \arg\min_{\pi}[-\mathbb{E}_{\tau\sim\pi}Q(s,a) + \beta\pmb{KL}(\pi, M_\pi)] \tag{7}$$

*Adding $\mathbb{E}_{\tau\sim\hat{\pi}_{\phi_m}}Q(s,a)$ in Equation 7, we can rewrite it as:*

$$\pi_m = \arg\min_{\pi}[\mathbb{E}_{\tau\sim\hat{\pi}_{\phi_m}}Q(s,a) - \mathbb{E}_{\tau\sim\pi}Q(s,a) + \beta\pmb{KL}(\pi, M_\pi)] \tag{8}$$

*Considering $\mathbb{E}_{\tau\sim\hat{\pi}_{\phi_m}}Q(s,a)$ is not related to the optimization objective, the above equation holds. Intuitively, we can observe that there exist two parts in the objective. About the first part, we can use importance sampling to obtain:*

$$\mathbb{E}_{\tau\sim\hat{\pi}_{\phi_m}}Q(s,a) - \mathbb{E}_{\tau\sim\pi}Q(s,a) = \mathbb{E}_{\tau\sim\hat{\pi}_{\phi_m}}[\frac{\hat{\pi}_{\phi_m} - \pi}{\hat{\pi}_{\phi_m}}Q(s,a)] \tag{9}$$

*Since $\hat{\pi}_{\phi_m}$ can be represented as $\arg\max_{\hat{\pi}_\phi}\mathbb{E}_{\tau\sim\hat{\pi}_\phi}[Q(s,a)]$ when $m$ approaching to infinity, the minimum of $\mathbb{E}_{\tau\sim\hat{\pi}_{\phi_m}}Q(s,a) - \mathbb{E}_{\tau\sim\pi}Q(s,a)$ can be achieved when the minimum of the following equation exits.*

$$\arg\min_{\pi}\|\hat{\pi}_{\phi_m} - \pi\| \iff \arg\min_{\pi}\|\arg\max_{\hat{\pi}_\phi}\mathbb{E}_{\tau\sim\hat{\pi}_\phi}[Q(s,a)] - \pi\| \quad as\ m\to\infty \tag{10}$$

*Let us see the second part in Equation 8. $\pi$ and $M_\pi$ are Gaussian distributions and the variances of distributions are constant in our framework. Thus, $\pmb{KL}(\pi, M_\pi) \iff \|\pi - M_\pi\|$ holds.*

*Hereafter, we can reformulate Equation 8 as follows:*

$$\pi_m = \arg\min_{\pi}[\|\arg\max_{\hat{\pi}_\phi}\mathbb{E}_{\tau\sim\hat{\pi}_\phi}[Q(s,a)] - \pi\| + \beta\|\pi - M_\pi\|] \tag{11}$$

*Based on the Lemma 1 in (Cheng et al., 2019), the solution to the above problem is derived as:*

$$\pi_m = \frac{1}{1+\beta}\hat{\pi}_{\phi_m} + \frac{\beta}{1+\beta}M_\pi \tag{12}$$

*To this end, the policy $\pi_m$ is the solution to the proposed optimization problem in this paper.*

**Theorem 2** *Let $D_{sub} = D_{TV}(\pi_{opt}, M_\pi)$ be the bias between the optimal policy and the prior policy, the policy bias $D_{TV}(\pi_m, \pi_{opt})$ in $m$-th iteration can be bounded as follows:*

$$D_{TV}(\pi_m, \pi_{opt}) \geq D_{sub} - \frac{1}{1+\beta}D_{TV}(\hat{\pi}_{\phi_m}, M_\pi)$$
$$D_{TV}(\pi_m, \pi_{opt}) \leq \frac{\beta}{1+\beta}D_{sub} \quad as\ m\to\infty \tag{13}$$

**Proof 2** *Note that the following derivation is most inspired by Theorem 1 in (Cheng et al., 2019). According to Lemma 1, the policy $\pi_m$ can be represented as $\frac{1}{1+\beta}\hat{\pi}_{\phi_m} + \frac{\beta}{1+\beta}M_\pi$.*

*Then, let us define the policy bias as $D_{TV}(\pi_m, \pi_{opt})$, and $D_{sub} = D_{TV}(\pi_{opt}, M_\pi)$. Since $D_{TV}$ is a metric that represents the total variational distance, we can use the triangle inequality to obtain:*

$$D_{TV}(\pi_m, \pi_{opt}) \geq D_{TV}(M_\pi, \pi_{opt}) - D_{TV}(M_\pi, \pi_m) \tag{14}$$

*According to the mixed policy definition in Equation 12, we can further decompose the term $D_{TV}(M_\pi, \pi_m)$:*

$$D_{TV}(M_\pi, \pi_m) = \sup_{(s,a)\in S\times A}\left|M_\pi - \frac{1}{1+\beta}\hat{\pi}_{\phi_m} - \frac{\beta}{1+\beta}M_\pi\right|$$
$$= \frac{1}{1+\beta}\sup_{(s,a)\in S\times A}|\hat{\pi}_{\phi_m} - M_\pi| \tag{15}$$
$$= \frac{1}{1+\beta}D_{TV}(\hat{\pi}_{\phi_m}, M_\pi)$$

*This holds for all $m \in \mathbb{N}$ from Equation 14 and Equation 15, and we can obtain the lower bound as follows:*

$$D_{TV}(\pi_m, \pi_{opt}) \geq D_{sub} - \frac{1}{1+\beta} D_{TV}(\hat{\pi}_{\phi_m}, M_\pi) \tag{16}$$

*The RL policy $\hat{\pi}_{\phi_m}$ can achieve asymptotic convergence to the (locally) optimal policy $\pi_{opt}$ through the policy gradient algorithm. In this case, we can derive the bias between the mixed policy $\pi_m$ and the optimal policy $\pi_{opt}$ as follows:*

$$
\begin{aligned}
D_{TV}(\pi_{opt}, \pi_m) &= \sup_{(s,a) \in S\text{x}A} \left| \pi_{opt} - \frac{1}{1+\beta}\hat{\pi}_{\phi_m} - \frac{\beta}{1+\beta}M_\pi \right| \\
&= \frac{\beta}{1+\beta} \sup_{(s,a) \in S\text{x}A} \left| \pi_{opt} - M_\pi \right| \qquad \text{as m} \to \infty \\
&= \frac{\beta}{1+\beta} D_{TV}(\pi_{opt}, M_\pi) \quad \text{as m} \to \infty \\
&= \frac{\beta}{1+\beta} D_{sub} \quad \text{as m} \to \infty
\end{aligned}
\tag{17}
$$

*Therefore, we obtain the upper bound.*

