# OpenReview forum: "Foundation Reinforcement Learning: towards Embodied Generalist Agents with Foundation Prior Assistance"
_ICLR.cc/2024/Conference — Submitted to ICLR 2024_

### Official Review · Reviewer_x7U5 · 2023-10-20

**Soundness:** 4 excellent
**Presentation:** 4 excellent
**Contribution:** 3 good
**Rating:** 8
**Confidence:** 5

**Summary:**

The paper proposes a new goal-conditioned reinforcement learning framework that leverages foundation models to accelerate learning, named Foundation Reinforcement Learning (FRL). FRL consists of three components, policy, value, and reward priors. To construct the policy prior, FRL utilize a pre-trained video generation model and an inverse dynamics model and distill them to a lightweight policy network. To construct the value prior, FRL uses VIP, which trains a universal value function on internet-scale robotic dataset, without fine-tuning. Finally, FRL uses 0-1 success function as the reward prior. FRL achieves superior performance compared to RL-only or foundation-only model methods on Meta-World.

**Strengths:**

- Novelty: The proposed framework that seamlessly integrates foundation models with reinforcement learning is indeed a novel contribution. Particularly striking is the concept of decoupling the policy and value priors, an unconventional move considering that values typically depend on policies, and utilizing the value prior as a potential-based reward function.
- Presentation: The meticulous ablation study presented in Section 5.3 adds a crucial layer of understanding by isolating and illustrating the distinct impact of each prior.
- Performance: Perhaps most compelling is its performance, exhibiting surprising sample efficiency in stark contrast to DrQ-v2.

**Weaknesses:**

- Limitation: While this paper is robust in its findings and methodology, acknowledging potential limitations would contribute to a more balanced discourse and pave the way for future research avenues. One such aspect is the comparison with methodologies like VLM that have a noted advantage in generating language-instructed policies.

**Questions:**

- Extension to multi-task RL: It seems the authors train a policy on each of individual tasks in Meta-World. However, an intriguing extension of this work would be exploring the feasibility of training a singular policy across multiple tasks

---

> ### Author Response · Authors · 2023-11-19
> **Response to Reviewer x7U5**
>
> Thank you for your comments and advice! We hope the following addresses your concerns:
>
> Weakness:
>
> > For the weakness, "acknowledging potential limitations would contribute to a more balanced discourse and pave the way for future research avenues.":
>
> Thank you for your suggestion. We agree that the limitations and future works should be discussed in the Discussion section. We have revised our paper in this section.
>
> Question:
>
> > For the question "Extension to multi-task RL: ...However, an intriguing extension of this work would be exploring the feasibility of training a singular policy across multiple tasks":
>
> Thank you for your comment. We think this question is very interesting. Currently, our distilled policy prior model is multi-task, but the agent we learned is not. We think that the policies learned in each environment can be distilled, and each policy can be regarded as a skill component conditioned on tasks. In this way, it can achieve multi-task learning in a better way. We will try this in the future.
>
> Finally, thanks again for your detailed suggestions! We have revised our paper and updated it on the website. And we highlight the changes and essential details that reviewers have mentioned in blue color.

---

> > ### Comment · Reviewer_x7U5 · 2023-11-21
> >
> > Thank you for the response. I have also read your response to the other reviewers and including more baselines (ALIX, TACO) would strengthen this paper. However, one question arises after reading the other reviewers' comments. What criteria did you use to choose the 8 tasks? Clearly explaining the criteria will better demonstrate that you did not cherry-pick the tasks and the proposed method will generalize to broader tasks.

---

> > > ### Author Response · Authors · 2023-11-21
> > > **Response to Reviewer x7U5**
> > >
> > > Thank you for your comments and suggestions. We hope the following addresses your concerns:
> > >
> > > The 8 tasks cover a variety of common scenarios in robotics, which are not cherry-picked. Instead, they are commonly used by other algorithms as they test different manipulation skills, including a broad spectrum of difficulties. For example, ROT [1], the SoTA IRL method, chooses 7 tasks in the meta-world: bin-picking, button-press-topdown, door-open, door-unlock, drawer-close, drawer-open, and hammer, which are included in our 8 tasks.
> > >
> > > We clarify this in the experimental settings in the paper. Thanks again for your suggestions.
> > >
> > > [1] Haldar, S., Mathur, V., Yarats, D., & Pinto, L. (2023, March). Watch and match: Supercharging imitation with regularized optimal transport. In *Conference on Robot Learning* (pp. 32-43). PMLR.

---

> > > > ### Comment · Reviewer_x7U5 · 2023-11-22
> > > >
> > > > Great! I really hope this paper gets accepted. I have increased the soundness from 3 to 4 and the confidence from 4 to 5.

---

> > > > > ### Author Response · Authors · 2023-11-22
> > > > > **Response to Reviewer x7U5**
> > > > >
> > > > > Thank you!!

---

### Official Review · Reviewer_AdQL · 2023-10-24

**Soundness:** 3 good
**Presentation:** 3 good
**Contribution:** 2 fair
**Rating:** 5
**Confidence:** 3

**Summary:**

This work proposes a foundation actor-critic framework which leverage foundation priors provided by foundation models. The expoeriments show the proposed method can learn efficiently and is robust to noisy priors with minimal human intervention.

**Strengths:**

1. This work investigates the adoptation of foundation models in the field of reinforement learning.
2. The proposed framework yields a promising performance on robotics manipulation tasks.

**Weaknesses:**

1. It looks like what happened in this paper is that the authors simply substituted the value and policy functions in the vanilla actor-critic with the VIP (Ma et al., 2022) model and the UniPi (Du et al., 2023) model, respectively. The so-called Reward Foundation Prior is still a 0-1 success signal as in the vanilla actor-critic framework. By doing so, the authors peddle a concept of foundation reinforcement learning. This manuscript looks like an empirical study of recent rl foundation models, without new theoretical contributions.
2. The experimental results mainly demonstrate that by simultaneously using Unipi and VIP, the proposed method learns faster than DrQ-v2 (ICLR 2022), while the proposed method is only compared (or comparable) to DrQ-v2, other methods either do not involve training (Unipi and Prior) or yield a success rate of 0% (R3M) in 6 out of 8 tasks. (See fig. 2). Also, experiments are insufficient. All the experiments are conducted in a simple environment. The proposed method is compared to a limited number of baselines. Experiments on more SOTA baselines and investigation of more foundation models are welcome.
2. From my perspective, I don't think the proposed method can be claimed as a brand new "Foundation Reinforcement Learning" framework since it simply uses two existing foundation methods.

**Questions:**

It would be appreciated if the authors could lay out more about the uniqueness of the proposed foundation actor-critic method to the vanilla ones except for the utilization of existing foundation models.

---

> ### Author Response · Authors · 2023-11-19
> **Response to Reviewer AdQL (1)**
>
> Thank you for your comments and advice! We hope the following addresses your concerns:
>
> > For the weakness, "The so-called Reward Foundation Prior is still a 0-1 success signal as in the vanilla actor-critic framework":
>
> Thank you for this comment. We agree that it is necessary to conduct experiments with success-reward prior. Considering there are few universal success-reward foundation models, we distill one proxy success-reward model with 50k data in total for all 8 tasks, which are labeled by the ground-truth success signals from replay buffers. The proxy model takes images as input and is conditioned on the task embeddings (multi-task), which has 1.7% false positive error and 9.9% false negative error on the evaluation datasets. Then, we run FAC with the three priors without any signals from the environment during training.
>
> The results are attached in Fig. 6 in App.A.1 (Page 14), which we will place in the main text later. We find that compared to receiving the ground-truth success-reward signals, FAC under the 50k-images-distilled model has a limited performance drop in the tasks generally. And it can achieve much superior performance than the FAC w.o. reward. Consequently, the proposed FAC can work well under the noisy success-reward foundation prior
>
> > For the weakness, "The proposed method is compared to a limited number of baselines. Experiments on more SOTA baselines and investigation of more foundation models are welcome.":
>
> We emphasize that the proposed Foundation Actor-Critic is based on a new setting, where all the signals for reinforcement learning come from the foundation models. Notably, there is no human manual-designed reward in the setting. Considering that there are few works targeting a similar setting, all the baselines are listed in Fig. 2. We also compare our method to the oracle setting with human-designed reward, and find that FAC achieves even better performance. It proves that we can use the foundation model in place of human-specified rewards for any new tasks.
>
> We also add some baselines under the same settings, where only the success-reward foundation prior is provided. We choose the recent SOTA model-free RL algorithms on MetaWorld ALIX [1] and TACO [2], as well as the baseline DrQ-v2 with the success-reward only. Notably, ALIX and TACO are both built on DrQ-v2. The results are shown in Fig. 7 in App.A.2 (Page 14-15), where '*' means that only 0-1 success reward is given.
>
> For your convenience, we summarize the results in the following table. Only FAC can achieve 100% success rates in all the environments. DrQ-v2*, ALIX*, TACO* can not work on hard tasks such as bin-picking and door-open. And FAC requires fewer environmental steps to reach 100% success rates, as shown in the Figure. The additional experiment results on the new baselines can also verify the significance and efficiency of utilizing the abundant prior knowledge for RL.
>
> | Success rate (%) with 3 runs | FAC  | DrQ-v2* | ALIX* | TACO* |
> | ---------------------------- | ---- | ------- | ----- | ----- |
> | bin-picking-v2               | 100  | 0       | 0     | 0     |
> | button-press-topdown-v2      | 100  | 0       | 0     | 0     |
> | door-open-v2                 | 100  | 0       | 0     | 0     |
> | door-unlock-v2               | 100  | 0       | 100   | 33.3  |
> | drawer-close-v2              | 100  | 100     | 110   | 100   |
> | drawer-open-v2               | 100  | 0       | 0     | 0     |
> | hammer-v2                    | 100  | 66.7    | 66.7  | 66.7  |
> | window-close-v2              | 100  | 100     | 33.3  | 66.7  |
>
> [1] Cetin, E., Ball, P. J., Roberts, S., & Celiktutan, O.. Stabilizing off-policy deep reinforcement learning from pixels. ICML 2022.
>
> [2] Zheng, R., Wang, X., Sun, Y., Ma, S., Zhao, J., Xu, H., ... & Huang, F.. TACO: Temporal Latent Action-Driven Contrastive Loss for Visual Reinforcement Learning. NeurIPS 2023.

---

> ### Author Response · Authors · 2023-11-19
> **Response to Reviewer AdQL (2)**
>
> > For the weakness, "This manuscript looks like an empirical study of recent rl foundation models, without new theoretical contributions...I don't think the proposed method can be claimed as a brand new 'Foundation Reinforcement Learning' framework since it simply uses two existing foundation methods.", and
> >
> > for the question, "It would be appreciated if the authors could lay out more about the uniqueness of the proposed foundation actor-critic method to the vanilla ones except for the utilization of existing foundation models.":
>
> The main target of our work is to propose the form of embodied prior and how to leverage them for RL. We do not train any new foundation models. But we emphasize that our framework is not limited to the model UniPi or VIP. Our proposed FRL and the FAC are agnostic to the foundation prior models. We just give an example of how to leverage the proxy foundation models to assist RL, and ultimately verify that our framework is effective. We have clarified in the paper that we mainly discuss the concrete form in which to represent embodied foundation priors. We have revised our story to be more accurate in paper. We think such a meta-level question we study is very significant because:
>
> Currently, there is a large amount of research on building foundation models for embodied AI, as we have mentioned in related works. However, these works are very different regarding the form of foundation models and are not even comparable. For example, R3M learns the visual backbone, and saycan learns task decomposer, while RT chooses VLM finetuning in an end-to-end way. They are studying to train the embodied foundation models from different perspectives, and the questions they try to answer are not the same.
>
> Instead, our work focuses on the concrete form in which to represent embodied foundation priors, rather than the actual RL algorithms that take advantage of the foundation priors. This is because there is no widely agreed embodied foundation model form that is widely accessible. And we think defining the form of the embodied foundation model is the first priority, which is the main contribution of this work. For example, in the BERT era, most researchers believed that BERT was the universal model, but GPT proposed to build large language models in an autoregressive way, which is another form to represent the language foundation prior knowledge. Similarly, what our paper discusses is exactly the form of the foundation priors in embodied AI.
>
> Finally, thanks again for your detailed suggestions! We have revised our paper and updated it on the website. And we highlight the changes and essential details that reviewers have mentioned in blue color.

---

> > ### Comment · Reviewer_AdQL · 2023-11-22
> >
> > Thanks for your answer! I agree with you that a concrete form of foundation priors in embodied AI is a fundamental problem. However, my main concern is that what actually happened in this paper seems a simple combination of existing value, policy, and reward foundation priors. I don't think that using them together can make the proposed approach a "foundation reinforcement learning" framework.

---

> > > ### Author Response · Authors · 2023-11-23
> > > **Response to Reviewer AdQL**
> > >
> > > Thank you for your response! We are glad that we have reached the consensus that a concrete form of foundation priors in embodied AI is a fundamental problem.
> > >
> > > We acknowledge that our foundation model is defined by the collection of policy, value, and success-reward foundation priors. However, **such concept of defining the priors as well as leveraging them has not previously been proposed**, and the study of their relative importance is lacking. We think that the merit of our method is to define how the foundation models in RL should be studied from a new perspective.
> > >
> > > Thanks again for your valuable review and suggestions! We are looking forward to your reply and are happy to answer any future questions.

---

> ### Author Response · Authors · 2023-11-21
> **Request of Reviewer's attention and feedback**
>
> Dear Reviewer **AdQL,**
>
> We kindly remind you that the rebuttal period is ending soon. Please let us know if our response has addressed your concerns.
>
> Following your suggestion, we have answered your concerns and improved the paper in the following aspects:
>
> - We conduct experiments with **success-reward prior** to demonstrate that the proposed FAC can work well under **the noisy success-reward foundation prior.**
> - We have **added 2 new baselines (ALIX, TACO)** **in** **all the 8 tasks** to demonstrate the advancement of our framework.
> - We have made **further clarification of our story as well as  the contributions**, and more explanations for the ablation experiments. For one thing, we emphasize that we target the concrete form in which to represent embodied foundation priors. For another, our framework is not limited to the model UniPi or VIP. Our proposed FRL and the FAC are **agnostic to the foundation prior models.**
>
> **All of these results and** **modifications** **have been included in the** revised paper.
>
> Thanks again for your valuable review. We are looking forward to your reply and are happy to answer any future questions.

---

### Official Review · Reviewer_rMFS · 2023-10-28

**Soundness:** 3 good
**Presentation:** 2 fair
**Contribution:** 3 good
**Rating:** 5
**Confidence:** 4

**Summary:**

The paper proposes to combine the benefits of pre-trained policy prior (from Unipi) and pre-trained value prior (from VIP) to improve reinforcement learning. The authors suggest learning a policy prior from Unipi and using it to regularize policy learning. They also use VIP as a value function prior and shaping reward.

While the paper's contribution is clear, the story may be confusing. If the authors aim to propose a framework for foundation RL, they should discuss 1. the concrete form in which to represent embodied foundation priors and 2. how to learn such a foundation model from the dataset. However, they didn't propose a new form for embodied foundation priors but leveraged existing embodied foundation priors(Unipi and VIP).

I think what the authors did, is to argue that both "Foundation Models for Policy Learning" and "Foundation Models for Representation Learning" in related works are important.  Unipi's policy prior is empirically weak due to a lack of interaction with the environment, while VIP's representation prior has not been fully leveraged. The authors propose novel ways to leverage VIP and combine the strengths of Unipi to achieve better performance.

**Strengths:**

The paper proposes to combine the benefits of pre-trained policy prior (from Unipi) and pre-trained value prior (from VIP) to improve reinforcement learning. Specifically, they propose novel ways to leverage VIP.

**Weaknesses:**

The story may be confusing. I think they did not really propose a novel framework of foundation RL, but propose novel ways to leverage and combine existing foundation models (UniPi and VIP).

However, if the authors can change the story to be more accurate, I will willing to raise my score.

**Questions:**

I am curious why you use R3M, rather than VIP, as the baseline for representation learning.

---

> ### Author Response · Authors · 2023-11-19
> **Response to Reviewer rMFS**
>
> Thank you for your comments and advice! We hope the following addresses your concerns:
>
> Weakness:
>
> > For the weakness, "The story may be confusing. I think they did not really propose a novel framework of foundation RL, but propose novel ways to leverage and combine existing foundation models (UniPi and VIP). The authors can change the story to be more accurate":
>
> Thank you for your suggestions.
>
> As you understand, our work is to propose the form of embodied prior and how to leverage them for RL. We do not train any new foundation models. But we emphasize that our framework is not limited to the model UniPi or VIP. Our proposed FRL and the FAC are agnostic to the foundation prior models. We just give an example of how to leverage the proxy foundation models to assist RL, and ultimately verify that our framework is effective. We call it a novel framework in the original paper because it is a paradigm shift. As you suggested, we have clarified in the paper that we mainly discuss the question (1) "the concrete form in which to represent embodied foundation priors", not (2) "how to learn such a foundation model from the dataset". We have revised our story to be more accurate in paper. We think such meta-level question (1) we study is quite significant because:
>
> Currently, there is a large amount of research on building foundation models for embodied AI, as we have mentioned in related works. However, these works are very different regarding the form of foundation models and are not even comparable. For example, R3M learns the visual backbone, and saycan learns task decomposer, while RT chooses VLM finetuning in an end-to-end way. They are studying to train the embodied foundation models from different perspectives, and the questions they try to answer are not the same.
>
> Instead, our work focuses on "the concrete form in which to represent embodied foundation priors", rather than the actual RL algorithms that take advantage of the foundation priors. This is because there is no widely agreed embodied foundation model form that is widely accessible. And we think defining the form of the embodied foundation model is the first priority, which is the main contribution of this work. For example, in the BERT era, most researchers believed that BERT was the universal model, but GPT proposed to build large language models in an autoregressive way, which is another form to represent the language foundation prior knowledge. Similarly, what our paper discusses is exactly the form of the foundation priors in embodied AI.
>
> Question:
>
> > For the question, "I am curious why you use R3M, rather than VIP, as the baseline for representation learning.":
>
> Thank you for this comment. We use R3M because it is the earliest universal visual backbone for robotics control tasks. As you have mentioned, we agree that it is necessary to train a baseline that takes the VIP as a visual backbone. We have updated the results in Figure 2. The results show that the model with VIP as the backbone (brown curve) fails in 5 environments with 0 success rate, which is slightly better than R3M (gray curve). Both of them have much poorer performance than FAC (green curve).
>
> Finally, thanks again for your detailed suggestions! We have revised our paper and updated it on the website. And we highlight the changes and essential details that reviewers have mentioned in blue color.

---

> ### Author Response · Authors · 2023-11-21
> **Request of Reviewer's attention and feedback**
>
> Dear Reviewer **rMFS,**
>
> We kindly remind you that the rebuttal period is ending soon. Please let us know if our response has addressed your concerns.
>
> Following your suggestion, we have answered your concerns and improved the paper in the following aspects:
>
> - We have made **further clarification of our story as well as  the contributions**, and more explanations for the ablation experiments. For one thing, we emphasize that we target the concrete form in which to represent embodied foundation priors. For another, our framework is not limited to the model UniPi or VIP. Our proposed FRL and the FAC are **agnostic to the foundation prior models.**
> - We have added **1 new baseline (VIP)** for representation prior, as you suggest. The model with VIP or R3M backbone has much poorer performance than FAC.
>
> **All of these results and** **modifications** **have been included in the** revised paper.
>
> Thanks again for your valuable review. We are looking forward to your reply and are happy to answer any future questions.

---

### Official Review · Reviewer_zJMD · 2023-10-29

**Soundness:** 2 fair
**Presentation:** 3 good
**Contribution:** 2 fair
**Rating:** 5
**Confidence:** 4

**Summary:**

The paper proposes a novel framework called Foundation Reinforcement Learning which leverages foundational priors: policy prior, value priod and success-reward prior which enables sample efficient training even under potentially noisy prior knowledge. The paper also proposes a foundational actor-critic algorithm that utilizes value, policy and success-reward prior knowledge. Paper shows empirical  results on 8 object manipulation tasks in Meta-World environment. FAC achieves 100% success on 7/8 tasks in less than 200k frames of training, significantly outperforming baselines like DrQv2 and R3M. In addition, paper also presents ablations to verify importance of each prior in the experimental setup.

**Strengths:**

1. Paper is well written
2. Idea of leveraging foundational priors to assist RL training and removing the need for human designed rewards is promising and interesting.
3. Preliminary evidence of policy working even with noisy foundation priors in MetaWorld environment is promising and shows initial signs of success of the proposed framework
4. Paper provides ablations to demonstrate importance of each component of the framework

**Weaknesses:**

1. Eventhough paper mention about using foundation priors for policy, value and success-reward the results use ground truth success-reward and do not show any results with foundation success-reward prior. Given the proposed contribution is a framework with foundation prior in policy, value and success-reward it is essential that authors present results with a foundation prior for success reward as well.
2. The proposed foundation actor critic algorithm is a simple extension to DrQv2 which just adds a KL constraint to the prior foundation policy. This idea of using a KL constraint is very common in RL finetuning literature where we have a access to pretrained policy (trained using demonstrations or any other prior data) that we want to finetune with RL.
3. The experimental setup is quite simple and doesn’t present extensive comparison with other baselines. For example, how does using a policy prior pretrained on a small dataset collected using a DRQv2 perform in comparison with the UniPi data? It is unclear if using data collected from a finetuned foundation model like UniPi is better than any other policy trained using in-domain data.
4. Comparison is only presented with DRQv2, R3M and other zero-shot baselines like UniPi and Prior. The experiment section needs to be expanded and compared with other commonly used baselines and SOTA methods on MetaWorld
5. Experiments are only presented in 1 simple environment where results on some tasks are not that convincing. For example, in figure 3 ablations results on button-press-topdown task shows that FAC vs FAC w.o. policy prior performs almost the same. Similarly, on door-open task FAC vs FAC w.o value prior performs almost the same. It is unclear from this experimental setup whether each component of the framework is important. Authors need to present experiments on more complex environments to clearly demonstrate importance of each component.
6. The core contribution of FRL is using foundation priors for RL training but in the experiments authors finetune the UniPi distilled policy with in-domain data which raises concerns that whether the policy prior benefits are coming purely from small in-domain finetuning or from base UniPi demonstrations that are used for distillation. It is essential that to disentangle these factors and present detailed ablations in the experiments.

**Questions:**

1. It’d be nice if authors can show benefits of foundation priors for policy without any finetuning on in-domain data for the tasks used in experiments as that is the most exciting result in my opinion.

I’d be happy to increase my score if authors address my concerns

---

> ### Author Response · Authors · 2023-11-19
> **Response to Reviewer zJMD (1)**
>
> Thank you for your comments and advice! We hope the following addresses your concerns:
>
> > For weakness 1, "it is essential that authors present results with a foundation prior for success reward as well.":
>
> Thank you for this comment. We agree that it is necessary to conduct experiments with success-reward prior. Considering there are few universal success-reward foundation models, we distill one proxy success-reward model with 50k data in total for all 8 tasks, which are labeled by the ground-truth success signals from replay buffers. The proxy model takes images as input and is conditioned on the task embeddings (multi-task), which has 1.7% false positive error and 9.9% false negative error on the evaluation datasets. Then, we run FAC with the three priors without any signals from the environment during training.
>
> The results are attached in Fig. 6 in App.A.1 (Page 14), which we will place in the main text later. We find that compared to receiving the ground-truth success-reward signals, FAC under the 50k-images-distilled model has a limited performance drop in the tasks generally. And it can achieve much superior performance than the FAC w.o. reward. Consequently, the proposed FAC can work well under the noisy success-reward foundation prior.
>
> > For weakness 2 "The proposed foundation actor critic algorithm is a simple extension to DrQv2 which just adds a KL constraint to the prior foundation policy.":
>
> Thank you for your comment. We acknowledge that such KL regularization term is simple and widely used in other algorithms.
>
> However, in this work, the main objective of this paper is to study the concrete form of the foundation models for embodied AI, rather than the actual RL algorithms that take advantage of the foundation priors. This is because there is no widely agreed embodied foundation model form that is widely accessible. And we think defining the form of the embodied foundation model is the first priority, which is the main contribution of this work. For example, in the BERT era, most researchers believed that BERT was the universal model, but GPT proposed to build large language models in an autoregressive way, which can be quite simple but effective. Similarly, our work aims to explore the concrete form of foundation priors for embodied AI. And we propose a novel learning paradigm for this, which does RL assisted by the policy, value and success-reward prior knowledge.
>
> It is an example of using KL regularization in DrQ-v2 regarding the policy prior, which is not the main contribution in our work. The FAC is a concrete and effective algorithm to demonstrate how to leverage the foundation priors.
>
> > For weakness 3 "how does using a policy prior pretrained on a small dataset collected using a DRQv2 perform in comparison with the UniPi data?",  and weakness 6 "whether the policy prior benefits are coming purely from small in-domain finetuning or from base UniPi demonstrations that are used for distillation.,
> >
> > and question "the authors can show benefits of foundation priors for policy without any finetuning on in-domain data for the tasks":
>
> These comments focus on how the policy prior works and why it requires in-domain data.
>
> As mentioned in the previous answer, our main contribution in this work is to investigate the concrete form of foundation priors for embodied AI. The answer in our paper is to use the foundation policy, value, and success reward model. In order to demonstrate that our framework is a valid paradigm of leveraging the foundation prior, we assume the policy prior can provide roughly correct actions (or even discretized directions). However, we find that the current policy foundation models can not work in the out-of-domain environments. (We include an example of the video prediction from the diffusion model without in-domain finetuning for your reference in the Supplementary Material). Therefore, that is why we use a small amount of data (10 videos for each task) to finetune the existing policy prior model to act as if this is the foundation prior for the general purpose. Our foundation RL framework is agnostic to any foundation prior models. However, different qualities of the foundation priors will yield different levels of guidance.
>
> We acknowledge that these foundation priors are not particularly ready. But the main target of this paper is to investigate what is the concrete form of the foundation priors for embodied AI and the importance of the priors. Hopefully, as more researchers agree with the significance of the proposed foundation priors, they will focus on how to build and improve the foundation prior (policy/value/success reward).

---

> ### Author Response · Authors · 2023-11-19
> **Response to Reviewer zJMD (2)**
>
> > For weakness 4 "The experiment section needs to be expanded and compared with other commonly used baselines and SOTA methods on MetaWorld":
>
> We emphasize that the proposed Foundation Actor-Critic is based on a new setting, where all the signals for reinforcement learning come from the foundation models. Notably, there is no human manual-designed reward in the setting. Considering that there are few works targeting a similar setting, all the baselines are listed in Fig. 2. We also compare our method to the oracle setting with human-designed reward, and find that FAC achieves even better performance. It gives the potential that we can use foundation model in place of human-specified reward for any new tasks.
>
> We also add some baselines under the same settings, where only the success-reward foundation prior is provided. We choose the recent SOTA model-free RL algorithms on MetaWorld ALIX [1] and TACO [2], as well as the baseline DrQ-v2 with the success-reward only. Notably, ALIX and TACO are both built on DrQ-v2. The results are shown in Fig. 7 in App.A.2 (Page 14-15), where '*' means that only 0-1 success reward is given.
>
> For your convenience, we summarize the results in the following table. Only FAC can achieve 100% success rates in all the environments. DrQ-v2*, ALIX*, TACO* can not work on hard tasks such as bin-picking and door-open. And FAC requires fewer environmental steps to reach 100% success rates, as shown in the Figure. The additional experiment results on the new baselines can also verify the significance and efficiency of utilizing the abundant prior knowledge for RL.
>
> | Success rate (%) with 3 runs | FAC  | DrQ-v2* | ALIX* | TACO* |
> | ---------------------------- | ---- | ------- | ----- | ----- |
> | bin-picking-v2               | 100  | 0       | 0     | 0     |
> | button-press-topdown-v2      | 100  | 0       | 0     | 0     |
> | door-open-v2                 | 100  | 0       | 0     | 0     |
> | door-unlock-v2               | 100  | 0       | 100   | 33.3  |
> | drawer-close-v2              | 100  | 100     | 110   | 100   |
> | drawer-open-v2               | 100  | 0       | 0     | 0     |
> | hammer-v2                    | 100  | 66.7    | 66.7  | 66.7  |
> | window-close-v2              | 100  | 100     | 33.3  | 66.7  |
>
> [1] Cetin, E., Ball, P. J., Roberts, S., & Celiktutan, O.. Stabilizing off-policy deep reinforcement learning from pixels. ICML 2022.
>
> [2] Zheng, R., Wang, X., Sun, Y., Ma, S., Zhao, J., Xu, H., ... & Huang, F.. TACO: Temporal Latent Action-Driven Contrastive Loss for Visual Reinforcement Learning. NeurIPS 2023.
>
> > For weakness 5 "...button-press-topdown task shows that FAC vs FAC w.o. policy prior performs almost the same. ...on door-open task FAC vs FAC w.o value prior performs almost the same. It is unclear from this experimental setup whether each component of the framework is important.":
>
> Thank you for your question. For most environments, it will be best to apply all the foundation priors. However, in some environments, either the policy prior or the value prior is accurate enough for solving the tasks, resulting in a few performance drops when removing the other prior. This depends on the quality of the foundation prior conditioned on the tasks. Generally, learning from the three embodied prior can be better than that from only two of them in the 8 environments. Here, we explain in detail for your mentioned cases.
>
> In button-press-topdown, the value prior can provide enough signals to guide the arm to press the button because the value is larger when the arm is closer to the button. And the policy prior can also provide a general direction. Thus, the agent can solve the tasks given only one prior (value or policy can both work) and can work faster given two priors together.
>
> In the door-open, the arm needs to approach the door first and then move away to open the door. In this process, the value prior provided by VIP becomes smaller when approaching the door and larger when opening it. Without policy prior, the arm has no signals to learn approaching the door since the value prior is lower when approaching the door. That is why FAC vs FAC w.o value prior performs almost the same, but FAC w.o. policy prior can not work.
>
> Finally, thanks again for your detailed suggestions! We have revised our paper and updated it on the website. And we highlight the changes and essential details that reviewers have mentioned in blue color.

---

> ### Author Response · Authors · 2023-11-21
> **Request of Reviewer's attention and feedback**
>
> Dear Reviewer **zJMD**,
>
> We kindly remind you that the rebuttal period is ending soon. Please let us know if our response has addressed your concerns.
>
> Following your suggestion, we have answered your concerns and improved the paper in the following aspects:
>
> - We conduct experiments with **success-reward prior** to demonstrate that the proposed FAC can work well under **the noisy success-reward foundation prior.**
> - We have **added 2 new baselines (ALIX, TACO)** **in** **all the 8 tasks** to demonstrate the advancement of our framework.
> - We have made **further clarification of our story as well as  the contributions**, and more explanations for the ablation experiments. For one thing, we emphasize that we target the concrete form in which to represent embodied foundation priors. For another, our framework is not limited to the model UniPi or VIP. Our proposed FRL and the FAC are **agnostic to the foundation prior models.**
>
> **All of these results and** **modifications** **have been included in the** revised paper.
>
> Thanks again for your valuable review. We are looking forward to your reply and are happy to answer any future questions.

---

> ### Comment · Reviewer_zJMD · 2023-11-22
>
> Thank you addressing some of my concerns. I have read authors response to all reviews.
>
> I agree with authors that their main contribution is a concrete form of foundation priors for embodied AI. However, to validate the proposed frameworks the experiments need to focus on diverse benchmarks, show results of meaningful generalization and be broadly applicable. The current results seem promising but are tested in limited tasks and it is unclear even with additional experiments that this framework is general enough to be applied in diverse set of tasks.
>
> Another concern I have is the proposed FAC framework is essentially saying that we should be used pretrained components for each of the policy, value and reward modules which is not a novel contribution. I don't think that using them together can make the proposed approach a "foundation reinforcement learning" framework.
>
>
> >We emphasize that the proposed Foundation Actor-Critic is based on a new setting, where all the signals for reinforcement learning come from the foundation models. Notably, there is no human manual-designed reward in the setting. Considering that there are few works targeting a similar setting, all the baselines are listed in Fig. 2. We also compare our method to the oracle setting with human-designed reward, and find that FAC achieves even better performance. It proves that we can use the foundation model in place of human-specified rewards for any new tasks.
>
> In response to W4 authors highlight this experiment, I would like to point out that authors shouldn't make strong claims like "It proves that we can use foundation model in place of human-specified reward for any new tasks" as these claims are backed by experiments on limited benchmarks.
>
> Based on the responses, I would like to thank authors to address weaknesses W1, W2, W3 and W5 through experiments and clarification. I recommend authors to incorporate this in the manuscript. I will update my rating accordingly.

---

> > ### Author Response · Authors · 2023-11-23
> > **Response to Reviewer zJMD**
> >
> > Thank you for your response! We are glad that some of your concerns are addressed. We hope the following can address the concerns in your new comments:
> >
> > > "Based on the responses, I would like to thank authors to address weaknesses W1, W2, W3 and W5 through experiments and clarification. I recommend authors to incorporate this in the manuscript"
> >
> > Thank you for your suggestions. We have incorporated the modifications towards the weaknesses W1, W2, W3, and W5, which **are highlighted in blue** in the manuscript.
> >
> > Specifically, for weakness w1, the results concerning the success-reward foundation prior are in App.A.1 (Page 14), which we will place **in the main text** **later**.
> >
> > The modifications towards weakness w2 are in
> >
> > For weakness w2 and w3, we have clarified the methodology in Sec. 1 (Page 2) and Sec. 4.1 (Page 5).
> >
> > For weakness w5, we have added more explanation and analysis in Sec. 5.3 (Page 8-9).
> >
> > We have revised our paper and updated it on the website for your reference.
> >
> > > "The current results seem promising but are tested in limited tasks and it is unclear even with additional experiments that this framework is general enough to be applied in diverse set of tasks."
> >
> > Thank you for your comment. The 8 tasks cover a variety of common scenarios in robotics, which are commonly used by other algorithms as they test different manipulation skills, including a broad spectrum of difficulties. For example, ROT [1], the SoTA IRL method, chooses 7 tasks in the meta-world: bin-picking, button-press-topdown, door-open, door-unlock, drawer-close, drawer-open, and hammer, which are included in our 8 tasks. Consequently, we think our framework is evaluated in a diverse set of tasks. We clarify this in the experimental settings in Sec. 5.1 (Page 7).
> >
> > Moreover, in response to your suggestion of more diverse benchmarks, we will conduct more experiments of our framework in more benchmarks in the future, due to the limited rebuttal time slot.
> >
> > [1] Haldar, S., Mathur, V., Yarats, D., & Pinto, L. (2023, March). Watch and match: Supercharging imitation with regularized optimal transport. In *Conference on Robot Learning* (pp. 32-43). PMLR.
> >
> > > "In response to W4 authors highlight this experiment, I would like to point out that authors shouldn't make strong claims like 'It proves that we can use foundation model in place of human-specified reward for any new tasks" as these claims are backed by experiments on limited benchmarks.'.
> >
> > Thank you for your suggestions. We have modified it to "It gives the potential that we can use foundation model in place of human-specified reward for any new tasks", which is also revised in the paper.
> >
> > Thanks again for your valuable review and suggestions! We are looking forward to your reply and are happy to answer any future questions.

---

### Meta-Review · Area_Chair_9VAA · 2023-12-15

**Metareview:**

The paper proposes to use foundation models as priors to an RL framework, thus making a 'foundation reinforcement learning' model. The model is tested on 8 tasks with good success. Most reviewers are critical because i) it is unclear whether using existing foundation models in an RL framework constitute a 'foundation RL model' (basically, is there overclaiming?), ii) if foundation models are indeed what brings the improvement given that tuning is also required, iii) if the experiments and settings chosen are 'too simple'. There is one reviewer that is particularly positive and champions the paper, however, the strengths noted by that reviewer are disputed by others (novelty?, results?). I will side with the critical reviewers that the story must become clearer, and that the current results make indeed a quite compelling case as early evidence, however, a more thorough manuscript and study is required before claiming 'Foundation Reinforcement Learning' in the title. If anything, the title would have to be 'Reinforcement Learning with Foundation Priors' or a variant.

**Justification For Why Not Higher Score:**

See above.

**Justification For Why Not Lower Score:**

See above.

---

### Decision · Program_Chairs · 2024-01-16

Reject